# Effect of a Plant-Based Nootropic Supplement on Perceptual Decision-Making and Brain Network Interdependencies: A Randomised, Double-Blinded, and Placebo-Controlled Study

**DOI:** 10.3390/brainsci15030226

**Published:** 2025-02-21

**Authors:** David O’Reilly, Joshua Bolam, Ioannis Delis, Andrea Utley

**Affiliations:** 1School of Biomedical Sciences, University of Leeds, Leeds LS2 9JT, UK; bolamj@tcd.ie (J.B.); i.delis@leeds.ac.uk (I.D.); a.utley@leeds.ac.uk (A.U.); 2Trinity College Institute of Neuroscience, Trinity College Dublin, D02 PN40 Dublin, Ireland

**Keywords:** nootropics, cognitive performance, information theory, EEG, brain health

## Abstract

Background: Natural nootropic compounds are evidenced to restore brain function in clinical and older populations and are purported to enhance cognitive abilities in healthy cohorts. This study aimed to provide neurocomputational insight into the discrepancies between the remarkable self-reports and growing interest in nootropics among healthy adults and the inconclusive performance-enhancing effects found in the literature. Methods: Towards this end, we devised a randomised, double-blinded, and placebo-controlled study where participants performed a visual categorisation task prior to and following 60 days of supplementation with a plant-based nootropic, while electroencephalographic (EEG) signals were concurrently captured. Results: We found that although no improvements in choice accuracy or reaction times were observed, the application of multivariate information-theoretic measures to the EEG source space showed broadband increases in similar and complementary interdependencies across brain networks of various spatial scales. These changes not only resulted in localised increases in the redundancy among brain network interactions but also more significant and widespread increases in synergy, especially within the delta frequency band. Conclusions: Our findings suggest that natural nootropics can improve overall brain network cohesion and energetic efficiency, computationally demonstrating the beneficial effects of natural nootropics on brain health. However, these effects could not be related to enhanced rapid perceptual decision-making performance in a healthy adult sample. Future research investigating these specific compounds as cognitive enhancers in healthy populations should focus on complex cognition in deliberative tasks (e.g., creativity, learning) and over longer supplementation durations. Clinical trials registration number: NCT06689644.

## 1. Introduction

Nootropics are a class of neurologically active compounds purported to enhance cognitive abilities such as memory, focus, and learning, hence the associated term, “smart drugs” [1]. Both synthetically derived (e.g., Ritalin, Piracetam, Modafinil) and naturally occurring compounds (e.g., caffeine, medicinal mushrooms) are taken by a growing proportion of the adult population for these performance-enhancing effects to aid with the demands of everyday modern life (e.g., rapid decision-making, creativity, productivity) [2,3,4,5]. Natural nootropics are considered to have less side effects and more potential widespread health benefits compared to synthetic compounds [4,6]; consequently, they have been of recent interest for their use in both clinical and healthy populations [1,7,8].

Among clinical human and animal studies, plant-derived extracts have been shown to improve neural and behavioural markers of health and performance, including perceptual motor function, learning, language comprehension, and memory [1,7,8,9,10,11]. The mechanisms by which these nootropics restore cognitive abilities varies widely but commonly involves increasing the supply and efficiency of use of energetic resources in the brain (e.g., increased cerebral blood flow, positive allosteric modulation of acetylcholine and glutamate receptors, and inhibition of monoamine oxidases) [8,12,13,14,15]. Additionally, they have been shown to promote neurogenesis, stimulate phospholipid metabolism in neurohormonal membranes, and enhance the beneficial cognitive effects of neurosteroids (e.g., pregnenolone sulphate, dehydroepiandrosterone sulphate) [12,16,17,18,19]. Hence, it is understandable that these substances would have impactful effects among cognitively impaired cohorts that typically present with reduced energetic resources or the efficiency of resource allocation (e.g., ageing, Alzheimer’s disease) [20,21,22]. Nonetheless, within the healthy working-age adult population, the interest in nootropic supplements has grown considerably recently for their purported ability to improve cognitive performance [2,3,12,23], with noteworthy self-reported accounts of their benefits [6,8,24]. However, it is currently unclear how well the impactful results from nootropic supplements found predominantly among clinical human and animal studies can be generalised to the healthy population as cognitive enhancers, as a significant number of the available studies in this group demonstrate minimally effective, contrasting, or even contradictory behavioural results [1,6,8,25,26,27,28,29]. To exemplify this point, nootropic effects have been observed to vary across age groups as a function of the underlying cognitive deficits associated with senescence [8,28,30], suggesting a restorative rather than enhancing effect of nootropic compounds [6,11]. Consequently, among high-functioning younger adults with adequate dietary regimes which likely function near their optimal state already, this “normalising” effect would be expected to be less pronounced, leading to more subtle changes in overall cognitive performance. Statistically speaking, the relatively small effect sizes from such subtle behavioural changes following supplementation likely explains the discrepancies in previous research; however, it does not explain the overwhelmingly positive self-reports and growing interest in these compounds among healthy adults [1,2,3,5,6,8,24]. These performance changes, although subtle in magnitude, may in fact be of considerable significance subjectively and may be underpinned by more pronounced effects neurologically. The energetic dynamics of the brain which nootropics likely affect can be cast through the mathematical lens of information theory [31,32], where information sharing across neuronal populations abstractly represents the electrophysiological dynamics of cognitive processes and, consequentially, the metabolic constraints underpinning them.

Therefore, in the current study, we aimed to investigate the efficacy of nootropic compounds in the healthy adult population to shed light mechanistically on these discrepancies in previous research and provide a neurocomputational explanation for the self-reports of improved wellbeing with nootropic consumption. Towards this objective, we devised a randomised, double-blinded, placebo-controlled experiment (pre-registration: https://osf.io/25afe (accessed on 6 July 2023)) that aimed to address this current research gap by combining behavioural experimentation with neuroscientific enquiry to answer the following research question:


*What effect does a commercially available nootropic supplement have on perceptual decision-making performance (i.e., the ability to make rapid decisions based on sensory information) and brain network interdependencies (i.e., the collective interactions between brain regions)?*


To answer this research question, we employed a computerised experimental paradigm testing participants’ visual perceptual decision-making while concurrently capturing brain activity using electroencephalography (EEG). We implemented this cognitive test pre- and post-60 days of daily supplementation with the Mind Lab Pro supplement. Our analytical approach consisted of applying multivariate information-theoretic measures to identify salient differences in brain network interdependencies that could be explained by the supplement regime. Prior to the experimental period, we hypothesised that this supplementation regime would result in negligible behavioural changes but more pronounced effects on the brain network interdependencies underlying perceptual decision-making, thereby explaining discrepancies in previous research and the growing interest and self-reports of improved wellbeing among healthy adults.

## 2. Materials and Methods

### 2.1. Participant Recruitment, Selection, and Randomisation Schedule

Participant recruitment strategies included poster advertisements in the local community, mass email notifications through approved university channels, and word of mouth. Potential candidates were provided with a brief outline of the experimental protocol and were asked to contact the experimental lead for further information and screening. Interested candidates were provided with an information sheet and were given time to ask questions before informed consent was obtained.

The eligibility screening of participants was guided by the following inclusion criteria: (i) be between 20 and 59 years old, (ii) be right-handed, and (iii) be able to cease taking other dietary supplements for two months. Exclusion criteria included the following: (i) currently consuming a nootropic supplement, (ii) have any known musculoskeletal or neurological medical conditions or cognitive impairments, (iii) have a known diagnosis of epilepsy/history of seizures, and (iv) have a known hearing or visual condition that affects daily life function.

Following successful screening, participants were pseudo randomly assigned on a 1:1 ratio basis to the treatment or control groups based on the order they were recruited into the study. Both the experimental lead and participants were blinded to the assignment of treatment and control groups. Only the principal investigator was aware of the group assignment but was not involved in experimental collection or analysis. After completing the experiment, each participant received an GBP 50 retail voucher (Amazon.com, Inc., Seattle, WA, USA). Following the completion of data collection and preliminary data analyses, both the experimental lead and participants were subsequently unblinded to group assignments.

Full ethical permission was gained from the Faculty of Biological Sciences ethics committee, University of Leeds (BIOSC22-022).

### 2.2. Interventional Compound

The treatment group received a commercially available Mind Lab Pro supplement (https://mindlabpro.com (accessed on 8 November 2024)), while the control group was given a matched placebo comprising an inactive cellulose substance. Participants of both groups were asked to consume two capsules per day (the lowest range of the recommended daily dose by the supplement manufacturer) continuously for 60 days, preferably in the morning with breakfast. Table 1 below provides a full list of the included ingredients for a two-capsule serving. They were asked to maintain their regular diet throughout supplementation and to immediately report any adverse effects.

### 2.3. Experimental Task

The day prior to and as soon as possible following the supplementation period, participants took part in a computerised cognitive task assessing their visual perceptual decision-making (Figure 1) [33]. In this experiment, a working memory task was also performed by the participants, which was not analysed in the current study. However, it is worth noting that the order of these separate experimental tasks was randomised across participants to prevent any carry-over effects (i.e., learning, fatigue).

### 2.4. Stimuli

The stimuli consisted of 18 face (Face Database; Max Plank Institute of Biological Cybernetics3) and 18 car (sourced from the Internet) grayscale images adapted from previous studies [33,34,35,36], retrieved to use as visual stimuli (image size: 512 × 512 pixels; bit depth: 8 bits/pixel) (Figure 1A). Each original image had its background removed before being transferred onto a uniform grey background (RGB: [115 115 115]), and images were equated for spatial contrast, frequency, luminance, and the total number of frontal and side views (i.e., a maximum of ±45°) to ensure identical magnitude spectra (i.e., average magnitude spectrum of all images in the database). All images had their corresponding phase spectra manipulated using the weighted mean phase technique, altering their image phase coherence and therefore characterising the phase coherence percentage and the quantity of visual sensory evidence available. Two levels of visual sensory evidence, 32.5% and 37.5% phase coherence, were selected to manipulate image classification difficulty (Figure 1B). A Stone 64-bit-based workstation (CPU: i7-9700; RAM 500 GB SSD) running Windows Professional 7 (Linux-x86_64-bit) and PsychoPy presentation software (v3.8.10) controlled the stimulus display (RGB: [128 128 128]). Images were presented on an Iiyama ProLite B2484HSU 24-inch monitor (Iiyama Corporation, Iiyama City, Japan) (resolution: 1920 × 1080 pixels; refresh rate: 75 Hz). Participants were positioned 60 cm from the monitor, and each image subtended approximately 8 × 8 degrees of visual angle.

### 2.5. Behavioural Task

Participants performed an object categorization task, in which they classified whether faces or cars were embedded in a series of images (Figure 1). Each trial began with a white (RGB: [255 255 255]) fixation cross presented on-screen for a randomised duration between 1 and 1.5 s. Then, a visual stimulus was presented for 50 ms. Participants were instructed to provide their response as soon as they had reached a decision (i.e., as quickly and accurately as they could), with a response deadline set to 1.25 s. Participants responded by pressing the correctly assigned key (i.e., left and right arrow presses for faces and cars, respectively) using their right index and middle fingers. Following an inter-stimulus interval (ISI; delay) of 100 ms, they then received visual feedback following each response for 500 ms. Three possible outcome statements, in block capitals, were presented as feedback, namely (1) correct (RGB: [0 255 0]), (2) incorrect (RGB: [255 0 0]), and (3) too slow (RGB: [0 0 255]), for correct, incorrect, and timed-out responses (i.e., exceeding the response deadline of 1.25 s), respectively. Single-trial reaction times (RTs) and choice accuracy (i.e., correct and incorrect) were collected as metrics of decision-making performance, with timed-out responses (i.e., exceeding the response deadline of 1.25 s) treated as incorrect. In total, participants completed 576 trials per session, consisting of four blocks of 144 trials each, with a 60 s rest period between blocks. Within each block, trials were divided equally between face and car images (imType) (i.e., 72 face and 72 car trials per block) and the two levels of stimulus phase coherence (imCoh) (i.e., 72 37.5% and 72 32.5% trials per block). The entire task lasted approximately 25–30 min.

### 2.6. EEG Recording and Pre-Processing

EEG signals were recorded in a sound-attenuated room using a 64-channel Brian Visions amplifier system and Analyzer software (Versions 2.1.1) at a sampling frequency of 1000 Hz. Following data capture, the EEG signals were processed in Matlab software R2023a using the EEGLab toolbox [37]. More specifically, following each data capture session, we firstly re-referenced the EEG signals to the average of all channels. We then bandpass-filtered the signals within the 0.5–200 Hz range. To remove power line noise, we subsequently applied spectrum interpolation (code taken from the Fieldtrip toolbox [38]) at 50 Hz and its corresponding harmonics up to 200 Hz [39]. To remove muscle and eye artefacts along with any remaining channel and line noise, we applied independent component analysis and, following an automated classification procedure, we removed artefactual components identified with a >90% certainty threshold. Stimulus-locked epochs (0–800 ms post-stimulus presentation) were extracted and averaged across trials for each participant to enhance the signal-to-noise ratio of the EEG signals. Finally, to enhance the spatial resolution of the EEG data for subsequent functional connectivity analyses in the EEG source space, we extracted the surface Laplacian for each participant’s data using a custom Matlab script [40].

### 2.7. Higher-Order Brain Network Interdependencies

Cognitive processes like perceptual decision-making involve complex interdependencies between different networks of neuronal populations operating across a range of frequency bands [41]. To provide a thorough mechanistic account of brain network interdependencies following nootropic supplementation, we therefore analysed the EEG signals from the representative stimulus-locked trial of each participant separately within the delta (0.5–4 Hz), theta (4–8 Hz), alpha (8–12 Hz), beta (12–30 Hz), and gamma (30–40 Hz) frequency bands and extracted brain networks at multiple scales to localise the putative nootropic effects. This delta band range was chosen even though the temporal resolution of the stimulus-locked trials was 1.25 Hz (i.e., 1/800 ms trial length = 1.25 Hz temporal resolution), as it was found in preliminary analyses that the partially represented cycles in the lower delta frequencies contributed significantly to the analysis output. Meanwhile, the gamma frequency range selected was based on the gamma frequency ranges available with EEG that are less prone to artefacts [42]. Temporal waveforms for each frequency band were extracted using a low- and high-pass bi-directional Butterworth filter combination with zero-phase distortion (“*filtfilt*” function in Matlab). To provide comprehensive insight into the changes induced by nootropic supplementation in the brain, we adopted a recently proposed greedy search algorithm (GSA) to identify higher-order networks of interdependencies between brain regions in a computationally efficient way that are maximally different between experimental groups [43]. More specifically, beginning with the representative stimulus-locked trial for each participant from the follow-up session, we firstly quantified the interdependence between all possible triplets of EEG channels. These interdependencies were quantified using two distinct information-theoretic measures of multivariate correlation strength, the total correlation (TC), and the dual total correlation (DTC) [44,45]. TC (Equation (1)) and DTC (Equation (2)) are non-negative multivariate generalisations of mutual information (MI), which for a system of n random variables denoted as $X^{n}=(X_{1},\ldots ,X_{n})$, can be expressed in terms of entropies (H) as (1)$$TC\left(X^{n}\right)=\sum_{j=1}^{n}H(X_{j})-H\left(X_{1},\ldots X_{n}\right)$$
(2)$$DTC\left(X^{n}\right)=H\left(X_{1},\ldots X_{n}\right)-\sum_{j=1}^{n}H\left(X_{j}|X_{-j}^{n}\right)$$

Here, for TC (Equation (1)), the sum of the Shannon entropies of individual variables in $X^{n}$ ($\sum_{j=1}^{n}H(X_{j})$) is contrasted against their joint Shannon entropy ($H\left(X_{1},\ldots X_{n}\right))$. Meanwhile, for DTC (Equation (2)), $H\left(X_{1},\ldots X_{n}\right)$ is contrasted against the sum of the conditional Shannon entropies ($\sum_{j=1}^{n}H\left(X_{j}|X_{-j}^{n}\right))$. The mathematical differences between these multivariate measures of MI results in the emphases of distinct types of collective interdependence. More specifically, TC quantifies the shared information in a system that is similar across variables (i.e., the collective constraints [46]), while DTC, also known as the binding information [47], quantifies the shared randomness across a system (i.e., complementary information) [46]. Hence, the subtraction of these two quantities (Equation (3)), a measure known as the O-Information (Ω) [46], quantifies the net balance between synergy and redundancy in a system (positive Ω values suggest predominantly redundant system interactions while negative values suggest net synergistic interactions). Through the application of TC, DTC, and Ω to networks of EEG source signals here, we aimed to provide important insight into the collective interactions of multiple brain regions and the types of statistical relationships they manifest, along with the changes induced in these characteristics with nootropic supplementation. In the current study, we used a Gaussian copula-based method to generate lower-bound estimates of TC and DTC and, consequently, Ω [48]. (3)Ω=TC−DTC

Returning to the GSA adopted from recent work [43], having quantified all pairwise interdependencies between EEG channels within a specific frequency band for each participant, we quantified the standardised mean difference for each pair between treatment and control groups using the Cohen’s d effect size (Equation (4)) [49]. Here, $\mu_{treatment}$ and $\mu_{control}$ are the average TC, DTC, or Ω values from the treatment and control groups, respectively, while s is the pooled standard deviation, where $s_{control}^{2}$ ($s_{treatment}^{2}$) and $m_{control}$ ($m_{treatment}$) are the variance and the sample size of the control (treatment) group, respectively. A positive (negative) value for d therefore represents the effects direction, indicating increases (decreases) in brain network interdependencies following the intervention. As both the positive and negative values of Ω were of interest in the current study, we reversed the signs of the estimated Ω values (i.e., -(TC−DTC)) and re-applied the GSA to identify maximally discriminative redundant and synergistic brain networks. (4)$$d=\frac{\mu_{treatment}-\mu_{control}}{s},\;\ldots s=\sqrt{\frac{\left(m-1\right)s_{treatment}^{2}+\left(m_{control}-1\right)s_{control}^{2}}{m_{treatment}+m-2}}$$

We then used this pairwise computation as the basis for subsequent computations at the higher interaction order of triplets by identifying the pair with the maximum (minimum) effect sizes and determining all possible triplet subsets that contain the identified pair of EEG channels. This procedure was further redeployed in a stepwise manner at successively higher interaction orders up to 16-channel networks, representing 1/4 of the entire scalp map and well within the range of reliable Ω estimation [43,50]. In doing so, we were able to identify large brain networks that maximally discriminated between experimental groups, both in the direction of greater and less network interdependencies among the treatment group compared to the control group while avoiding the combinatorial explosion inherent to computing higher-order interactions.

Having identified the brain networks most different between groups at the follow-up session across multiple interaction orders, be they significantly higher or lower than the control group, we then worked backwards to compute the same brain networks found to be significantly discriminative at follow-up but from the pre-session EEG data. These baseline values were then included as a covariate in separate analysis of covariance (ANCOVA) models for each frequency band and interaction order alongside experimental group affiliation as a fixed factor and the follow-up session TC or DTC values as the dependent variable (SPSS Statistics 28 software). This enabled us to control for baseline differences to effectively ascertain whether these group differences at follow-up are genuine nootropic effects or could be simply explained by differences present from the outset of the experiment.

### 2.8. Statistical Analyses

For behavioural analyses, trials where participants responded <300 ms or >1200 ms post-stimulus presentation were discarded as “*fast guesses*” and “*attentional lapses*”, respectively [51]. The median RT for each participant at baseline and follow-up was calculated from the correct trials only, while the percentage of correct total trials was also taken to summarise choice accuracy. These summary statistics were employed in Mann–Whitney U tests to determine statistical differences between groups. Furthermore, to determine if specific features of the perceptual decision-making task were influenced by the nootropic, the choice speed and accuracies were also summarised in the same way but from easy and hard trials (i.e., high and low imCoh, respectively) and the different imType trials (i.e., face and car images) only. To ensure our results are not limited by a sample size, we also capitalised on the large number of trials of reaction time data to test for statistical differences between pre- and post-sessions at the within-participant level using a Bayesian prevalence approach [52]. Specifically, test statistics were generated for each participant, and group-level statistical differences were inferred through Monte Carlo simulations (N = 10,000 runs). Statistical significance was set a priori to *p* < 0.05. False discovery rates (FDRs) during the ANCOVA procedures (see “Higher-order brain network interdependencies” of the Section 2) were controlled separately for models testing greater than (i.e., maximal Cohen’s d) and less than (i.e., minimal Cohen’s d) brain network interdependencies using the Benjamini and Hochberg approach [53]. All ANCOVA models met the assumption of homoskedasticity as per White’s test (*p* > 0.05).

## 3. Results

In total, 37 participants (treatment group = 19 participants; control group = 18 participants) were successfully recruited; however, five participants (treatment group = 2 participants; control group = 3 participants) voluntarily withdrew before the supplementation period ended, leaving 32 participants (treatment group = 17 participants; control group = 15 participants) as the study sample. No significant differences between groups were present for age (treatment group: 38 ± 8.3; control group: 31.7 ± 10.3 (*p* > 0.05)), while a relatively even gender split was found in both groups (treatment group: M = 10/F = 7; control group: M = 8/F = 7). Of the two participants from the treatment group that withdrew, neither self-reported adverse effects, demonstrating an overall good tolerance for the supplement across the cohort. Participants were assessed 3.66 ± 3.2 days after their scheduled supplementation period was complete and self-reported good adherence to the supplementation regime.

### 3.1. Nootropic Supplementation Did Not Improve Perceptual Decision-Making Performance

We found no statistically significant differences within or between experimental groups (*p* > 0.05) across correct trials of any image type or difficulty level (“*Correct trials*”, Figure 2(1A,2A)), high imCoh trials (“*Easy trials*”, Figure 2(1B,2B)), low imCoh trials (“*Hard trials*”, Figure 2(1C,2C)), car imType trials (“*Car trials*”, Figure 2(1D,2D)), or face imType trials (“*Face trials*”, Figure 2(1E,2E)) for choice speed (Figure 2(1)) or choice accuracy (Figure 2(2)). Both groups typically demonstrated slower reaction times at follow-up compared to baseline; however, this increase was lower in the treatment group. This resulted in the treatment group demonstrating typically faster reaction times at follow-up compared to controls, except for the hard trials (see Figure 2(1A,B,D,E)). However, no evidence was found for an improvement in choice reaction times due to nootropic supplementation. Further testing of these differences at the within-participant level using a Bayesian prevalence approach revealed that no group-level effects could be inferred (i.e., the confidence intervals from the Monte Carlo simulations fell below zero), with a minority of both groups (treatment = 7; control = 5) demonstrating significant reductions in choice reaction times.

For choice accuracies, both groups scored highly at both sessions (>80% of trial correct on average), suggesting the task was well within their capabilities. The control group demonstrated a slight improvement in the percentage of trials correct at follow-up compared to baseline, a trend that was generalised across all imType and imCoh trials. This coincides with the subtle increase in reaction times in this same group, suggesting a change in the speed and accuracy bias across the intervention in this group. Meanwhile, the treatment group demonstrated no noticeable changes in choice accuracy following the intervention and had typically lower choice accuracy than the control group.

### 3.2. Information Sharing Across Brain Networks Is Enhanced Following Nootropic Supplementation

The application of TC and DTC as part of a GSA to the representative stimulus-locked trials of participants identified maximally discriminative brain networks across a range of interaction orders and frequency ranges (Figure 3 and Figure 4). All the identified brain network interdependencies quantified using TC and DTC were in favour of enhanced information sharing among the treatment group at follow-up compared to controls, controlling for baseline differences. No brain networks exhibiting greater information sharing among the control group were found for TC or DTC; therefore, results in this direction are not illustrated here.

Beginning with the findings from TC (Figure 3) following FDR correction (critical value for *p* = 0.031), significantly greater network interdependencies were found in all frequency bands except the gamma band (Figure 3A). Typical TC value ranges within baseline and follow-up sessions for both groups (control = red; treatment = blue) for all significant frequency bands are illustrated in Figure 3B.1,B.2, respectively. The most discriminative networks from the significant frequency bands were of a relatively low interaction order (see Figure 3C for the scalp topography with the most discriminative brain networks highlighted). The delta band demonstrated significantly enhanced TC among treatment group participants up to the sixth order (Figure 3A); however, the fourth-order brain network was most discriminative here (F = 5.61, (*p* = 0.0152)). This brain network consisted of EEG sources covering the left and right frontotemporal and the centroparietal region (Figure 3C). Meanwhile, the theta, alpha, and beta bands all demonstrated significantly enhanced information sharing for brain networks up to the 13th order for theta and 16th order for both alpha and beta bands (Figure 3A). However, the most discriminative brain networks for all three frequencies were consistently composed of triplets of EEG sources (theta: F = 9.1 (*p* = 0.0059); alpha: F = 12.95 (*p* = 0.0014); beta: F = 14.5 (*p* < 0.001)) (Figure 3A). For the theta band, this triplet consisted of two EEG sources over the central lobe and one on the right temporal lobe (Figure 3C), while the alpha band triplet consisted of EEG sources originating from the left temporal lobe only. Finally, for the beta band, which demonstrated the most significant difference between groups at follow-up controlling for baseline differences, the triplet was composed of more widespread EEG sources from the frontal, right temporal, and parietal brain regions.

The nootropics effect of information sharing across brain networks was much more widespread and significant when quantified using DTC (Figure 4A–C), suggesting the nootropic enhanced the sharing of complementary information more so than redundant information. All five frequency bands here displayed significantly enhanced information sharing (FDR critical value: *p* = 0.0027), but in contrast to TC (Figure 3), these differences steadily increased for the most part as a function of the interaction order (Figure 4A). This trend was especially obvious for the beta band, where the highest-order network analysed was also the most discriminative (F = 63.04 (*p* < 0.00001)) (Figure 4A). Interdependencies between 12 EEG sources within the delta band demonstrated the most significant group differences (F = 100.41 (*p* < 0.00001)), followed by 12th-order network interdependencies within the alpha (F = 65.98 (*p* < 0.00001)) and theta (F = 63.76 (*p* < 0.00001)) bands. The gamma band was noticeably lower in terms of its group differences, with a 13th-order network of EEG sources demonstrating the highest discrimination in this frequency range (F = 42.4 (*p* < 0.00001)). Based on these results, it is likely that significant enhancements to information sharing would be found in higher-order brain networks than those analysed here. As found in the TC results (Figure 3B.1,B.2), the alpha band provided the greatest amount of shared information across the intervention, while the delta band typically provided the least bits of information (the bits of information were normalised by the interaction order for comparability) (Figure 4B.1,B.2).

### 3.3. Natural Nootropic Supplement Increases Both the Redundancy and Synergy Between Brain Regions

The balance between redundant (i.e., informationally similar) and synergistic (i.e., informationally complementary) interdependencies in the brain pre- and post-supplementation with a nootropic or placebo was quantified using the O-Information (Ω), revealing salient group differences across multiple interaction orders and frequency bands that all favoured the treatment group in terms of increased redundancy and synergy (Figure 5 and Figure 6). As with the TC and DTC results (Figure 3 and Figure 4), no significant differences were found in favour of greater redundancy or synergy among the control group; hence, findings favouring the treatment group only are illustrated here. We found that the discriminative networks of EEG sources identified were redundancy-dominated (i.e., positive Ω values) across sessions and groups for both the redundancy (Figure 5) and synergy (Figure 6) analyses. Therefore, the boxplots in Figure 6B.1,B.2 of the synergy analysis depict a significant decrease in redundant information (corresponding to an increase in synergistic information) for the treatment group (blue boxes) compared to controls (red boxes) within the delta and gamma frequency bands.

Significantly increased redundancy was found in all frequency bands following FDR correction (critical value: *p* = 0.0278), where three frequency bands (i.e., delta, alpha, and beta) were shown to comprise significantly different redundancies for network interaction orders up to the maximum analysed here (i.e., 16 channels) (Figure 5A). Meanwhile, the theta and gamma bands demonstrated a much lower prevalence for this nootropic effect, demonstrating significantly greater redundancy up to the sixth and third interaction orders, respectively (Figure 5A). As found in the TC results (Figure 3), these differences were most significant at lower interaction orders, suggesting local information processing at specific brain regions was enhanced post-supplementation. Indeed, the delta band was the only frequency range where the most significant network was composed of more than three EEG sources, with the sixth order being most significant (F = 19.6 (*p* < 0.001)). This band also demonstrated the largest amount of redundant information across sessions and groups (displayed in normalised bits (Figure 5B.1,B.2) and incorporated left frontotemporal and right temporoparietal regions (Figure 5C). The beta band triplet was the most different between groups (F= 19.76 (*p* < 0.001)) and included occipital and right temporal–occipital brain regions (Figure 5C).

The significant reductions in redundancy (increases in synergy) found among the treatment group were frequency band-specific, with the predominant effects found in the delta band followed by the gamma band (FDR critical value: *p* = 0.0101) (Figure 6A). All other frequency bands did not demonstrate any nootropic effects. The delta band demonstrated its most significantly different network interdependencies at the seventh interaction order (F = 25.1 (*p* < 0.0001)); however, these significant differences continued in a mostly consistent way up to the maximum 16th order analysed here (Figure 6A). Indeed, this effect was more pronounced than that found of the statistical differences found within the delta band or any other band among the maximally redundant networks (Figure 5), suggesting an overall effect in the direction of increased synergy for the delta band. This most significantly different brain network in the delta band was composed of EEG sources scattered across frontal, temporal, and parietal brain regions (Figure 6C). Meanwhile, for the gamma band, the fifth-order network was most significantly different (F = 9.9 (*p* = 0.0043)) and was more focussed around the frontal and left temporal brain regions (Figure 6C).

## 4. Discussion

This randomised, double-blinded, placebo-controlled study investigated the effects of a plant-based nootropic supplement on visual perceptual decision-making performance and brain network interdependencies in a healthy adult cohort. Through a visual categorisation task where participants performed consecutive trials of rapid perceptual decision-making based on visual stimuli (i.e., face vs. car images) while EEG signals were concurrently captured, we analysed changes in perceptual decision-making performance (i.e., choice accuracy and reaction times) and the underlying brain network interdependencies following 60 days of nootropic supplementation. Supporting our a priori hypothesis, we found evidence for pronounced neurophysiological changes despite no significant behavioural improvements (i.e., choice accuracy or choice reaction times). Specifically, we found broadband changes in brain network interdependencies of various interaction orders that suggest nootropic supplementation increased the strength of complementary and similar statistical dependencies between brain regions, resulting in an overall enhancement of brain network cohesion and computational capacity. The findings presented here offer a neurocomputational explanation for the increased interest in and positive self-reports of using nootropic supplements in healthy adult cohorts despite the inconclusive behavioural effects found in the literature.

Through the application of multivariate measures of statistical dependency (i.e., TC and DTC) within a greedy search algorithm [28], we consistently identified significantly greater sharing of both similar and complementary information between EEG sources among the treatment group (see Figure 3 and Figure 4). Remarkably, these significantly different brain networks were found across a range of spatial scales (k = 3–16 channels) and frequency bands (i.e., delta–gamma ranges) and, through the application of the O-Information [31,35], were shown to signify an overall shift towards increased synergy in the brain following nootropic supplementation (see the greater F-statistics in Figure 4A and Figure 6A compared to Figure 3A and Figure 5A). Although the brain networks analysed here were strongly redundancy-dominated (see Figure 6B.1,B.2) and redundancy was shown to increase across several localised EEG sources (Figure 3 and Figure 5), the most significant changes from baseline were found in the direction of increased complementarity (Figure 4) and reductions in redundancy dominance (i.e., increased synergy) (Figure 6), together suggesting improvements in both local and distributed information processing [54]. This also suggests that the crucial balance between redundancy and synergy as functionally segregative and integrative forces, respectively, in dynamic systems like the brain was maintained following nootropic supplementation, thus ensuring adequate robustness (through compensatory increases in redundancy) to support the overall increase in computational capacity gained with increased synergy [54,55,56]. Indeed, the prevalence of synergistic interactions has been closely linked to goal-directed learning [57], the evolution of human intelligence and different states of consciousness [58,59], and it contributes to metabolically efficient neural codes [60,61]. Hence, the main result of this study is of crucial evidentiary importance towards demonstrating the efficacy of nootropic supplements in supporting brain health in the general adult population.

However, aside from the evidence provided here for nootropic supplements supporting brain health, further work is required to fully understand their efficacy as cognitive performance enhancers. Characteristics of the task such as the limited sample size restrict the conclusions that can be made from this study in this regard. The participants of both groups exhibited high choice accuracies at both sessions, which could be perceived as the task not being sufficiently difficult to tease out the treatment effects. However, this low error rate provided us with a large number of valid trials to probe the brain networks underpinning perceptual decision-making, which ultimately lead to identifying a consistent enhancement of functional complementary across brain regions in the treatment group. Regarding sample size, our study is similar to most of the related literature that has frequently demonstrated subtle behavioural effects [6,12,25,27,28,29], and our behavioural results were further supported by Bayesian inference on within-participant tests performed on the large-scale RT trial data [52]. This contrasts with other nootropic compounds renowned for their obvious cognitively stimulating effects (e.g., caffeine) [28,62] but which result in withdrawal symptoms not found among the compounds analysed here, likely due to the modulatory effect on cortical energy expenditure with caffeine consumption [62]. The compounds analysed here perhaps fall under a separate class of nootropics with more subtle effects on brain metabolism and function among healthy adults that manifest over longer timescales [1,8]. Furthermore, the supplements’ predominant effect on complementary cortical interdependencies in the delta frequency range (Figure 6) closely aligns with recent findings demonstrating global increases in synergistic brain interdependencies during meditation in the same neural oscillations [63]. Hence, our conjecture here is that research on these nootropic compounds as performance enhancers among healthy adults should instead follow a similar vein as studies on meditation and mindfulness practises that consistently found significant effects on complex cognition (e.g., creativity, learning) [64,65]. This position is supported by the multimodal integrative functions of synergy-dominated cortical regions most likely effected by the nootropic; however [55,59], due to the broadband prevalence of these effects among redundant brain interdependencies (Figure 5), it does not limit the potential of these compounds in other domains [66,67]. Future research on these compounds in the healthy adult population should therefore examine their effects using complementary modalities (e.g., fMRI) across greater supplementation durations and in more deliberate cognitive tasks.

## 5. Limitations

The conclusions drawn from this study’s behavioural findings are limited by the sample size included; although within-participant statistical testing supported our findings and the fact the numbers are in line with other studies, this should still be considered. This study suggests that the behavioural effects are subtle in comparison to the neurological effects of the nootropic supplement in healthy adults but does not rule out the subjective significance of any induced changes in cognitive performance, however small. Moreover, we found anecdotally that the participants recruited had a disproportionately high level of educational attainment compared to the general population, suggesting their cognitive function and so may not be fully representative of the general population. Nonetheless, the cognitive load required for this task was low, and previous studies employing this experimental paradigm have found it to be generally applicable to various populations (e.g., young versus old) [34]. The restricted temporal resolution of the stimulus-locked trials analysed here (i.e., 1.25 Hz) resulted in the whole period of lower delta band frequencies not being fully represented. Nevertheless, removal of this lower delta range (0.5–1.25 Hz) in preliminary analyses resulted in the correlation strength of many network interdependencies illustrated here being lost. Hence, we suggest that the partial coverage of these lower frequency ranges given in this analysis provided crucial insight into the nootropic’s effects, as evidenced by the statistically significant effects persistently shown (see Figure 3, Figure 4, Figure 5 and Figure 6). Future work with longer naturalistic stimulations will most likely demonstrate even more prominent nootropic effects within this delta frequency range.

## 6. Conclusions

This study provides neurocomputational insight into the discrepancy between the growing popularity of natural nootropic supplements in the healthy adult population and their inconclusive cognitive-enhancing effects, showing that pronounced neurophysiological effects can occur in the absence of behavioural performance changes. Our findings support the idea that natural nootropic compounds can improve brain health by enhancing the synergy between brain regions, but these benefits do not necessarily result in cognitive enhancement among healthy working-age adults, specifically in rapid visual perceptual decision-making. Future work should focus on more deliberative cognitive tasks and use complementary experimental modalities to comprehensively explain mechanistically the cognitive effects of natural nootropics.

## Figures and Tables

**Figure 1 brainsci-15-00226-f001:**
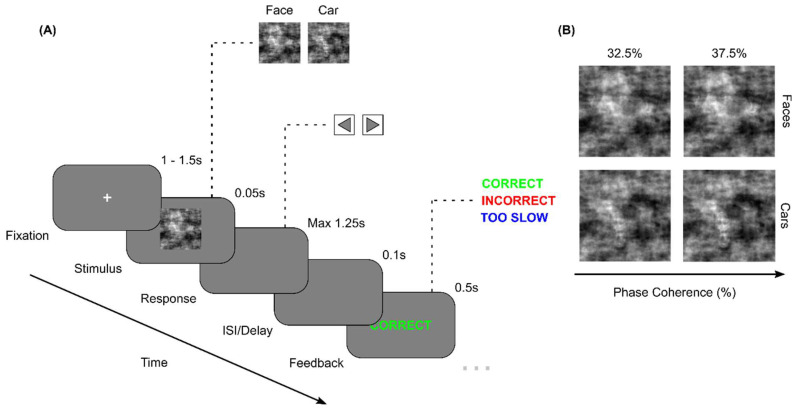
(**A**) Schematic representation of the experimental paradigm on a single-trial basis. Participants were instructed to classify noisy images of faces and cars, presented for 50 ms, by indicating their response with left and right key presses, respectively, within a 1.25 s deadline following stimulus presentation. Following an inter-stimulus interval (ISI; delay) of 100 ms, feedback was presented for 500 ms (correct, incorrect, or too slow in block capitals). (**B**) Sample face (top row) and car (bottom row) images at the two levels of visual phase coherence (i.e., 32.5% and 37.5%) used in the task.

**Figure 2 brainsci-15-00226-f002:**
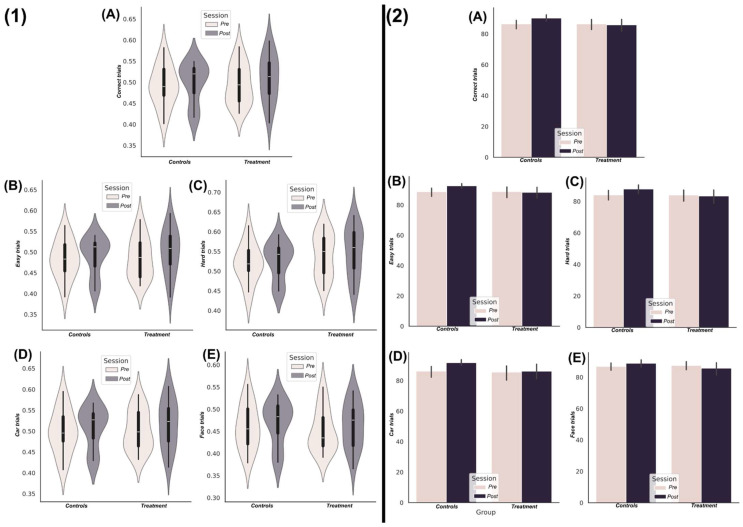
(**1**) The choice reaction times (ms) at the pre- (light colour) and post-session (dark colour) of treatment and control groups for correct trials (**A**), easy trials (**B**), hard trials (**C**), car trials (**D**), and face trials (**E**) depicted as violin plots to illustrate the distribution of reaction times. (**2**) In the same order of trial types (**A**–**E**), the choice accuracies are displayed as the median and standard deviation of the percentage of total trials correct for both experimental groups at the pre- (light colour) and post- (dark colour) sessions.

**Figure 3 brainsci-15-00226-f003:**
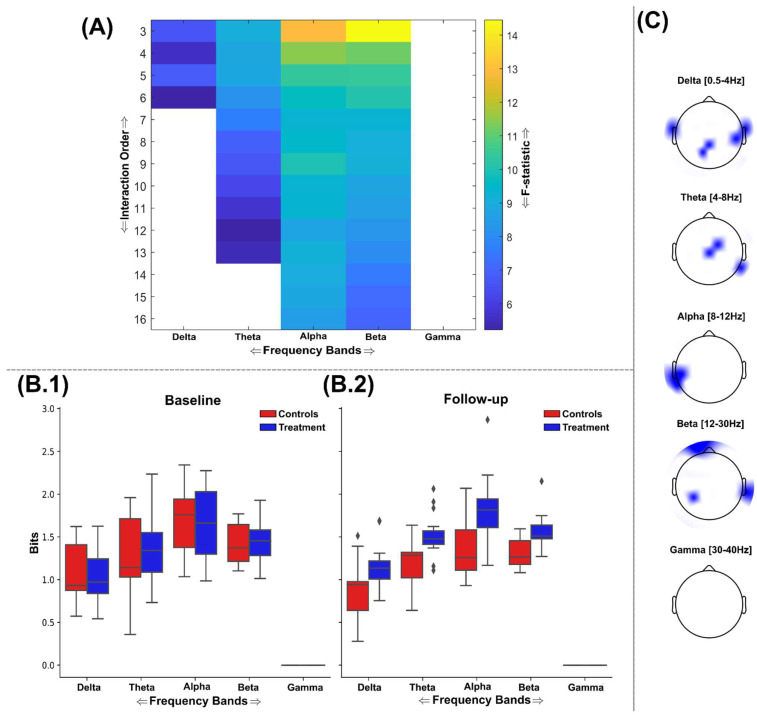
An overview of the findings from the application of TC to the EEG source signals using the GSA. (**A**) The F-statistic representing the group differences for each frequency band and interaction order taken from ANCOVA models that controlled for baseline differences in TC values. No significant differences in favour of greater TC among the control group were found, so the treatment group results are illustrated only. Areas coloured white indicate no significant differences were found. The interquartile ranges of the most significant interaction order TC values for control (red) and treatment (blue) groups at the baseline (**B.1**) and follow-up (**B.2**) sessions are depicted as boxplots for each frequency band. The TC values illustrated were normalised by the interaction order for comparability. The gamma band contained no significantly different network dependencies and so was not illustrated here. (**C**) The most significantly different brain networks were also illustrated topographically, where blue-shaded areas highlighted the included EEG sources.

**Figure 4 brainsci-15-00226-f004:**
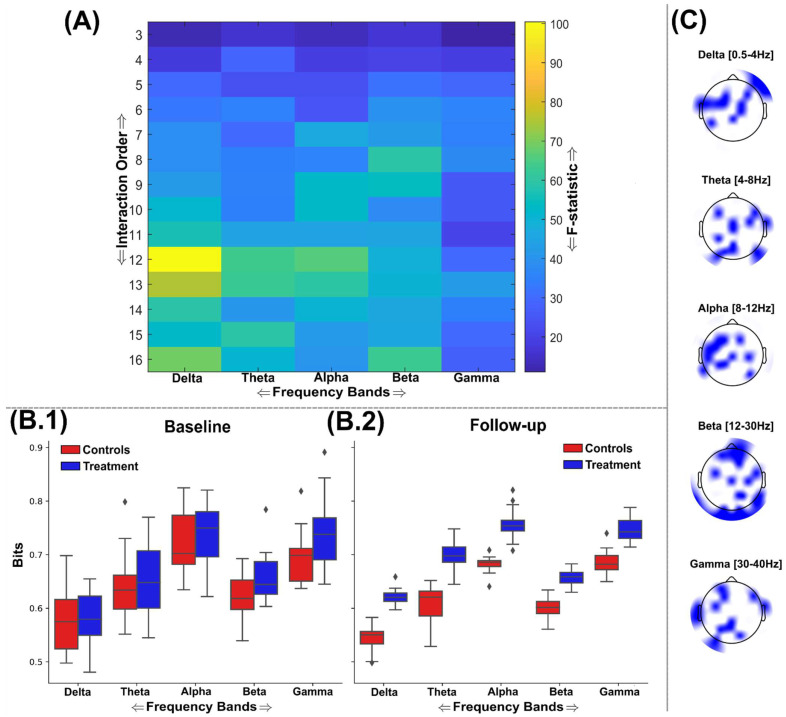
An overview of the findings from the application of DTC to the EEG source signals using the GSA. (**A**) The F-statistic representing the group differences for each frequency band and interaction order taken from ANCOVA models that controlled for baseline differences in TC values. No significant differences in favour of greater DTC among the control group were found, so the treatment group results are illustrated only. The interquartile ranges of the most significantly different network DTC values for control (red) and treatment (blue) groups at the baseline (**B.1**) and follow-up (**B.2**) sessions are depicted as boxplots for each frequency band. The DTC values illustrated were normalised by the interaction order for comparability. (**C**) The most significantly different brain networks were also illustrated topographically, where blue-shaded areas highlighted the included EEG sources.

**Figure 5 brainsci-15-00226-f005:**
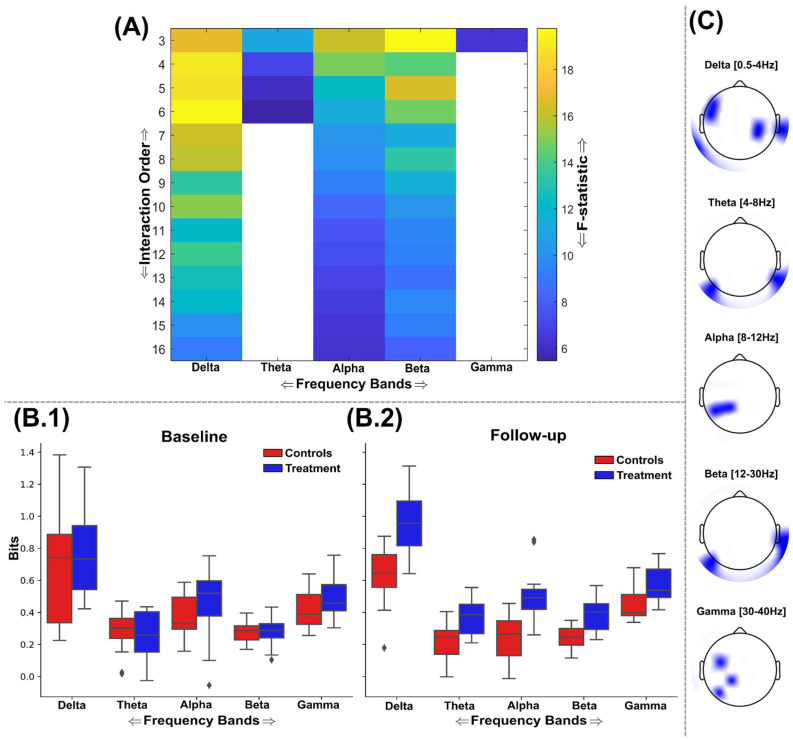
An overview of the findings from the application of O-Information (Ω) to the EEG source signals using the GSA to find maximally discriminative networks of redundant interactions. (**A**) The F-statistic representing the group differences for each frequency band and interaction order taken from ANCOVA models that controlled for baseline differences in Ω values. No significant differences in favour of greater Ω among the control group were found, so the treatment group results are illustrated only. Areas coloured white indicate no significant differences were found. The interquartile ranges of the most significant interaction order Ω values for control (red) and treatment (blue) groups at the baseline (**B.1**) and follow-up (**B.2**) sessions are depicted as boxplots for each frequency band. The Ω values illustrated were normalised by the interaction order for comparability. (**C**) The most significantly different brain networks were also illustrated topographically, where blue-shaded areas highlighted the included EEG sources.

**Figure 6 brainsci-15-00226-f006:**
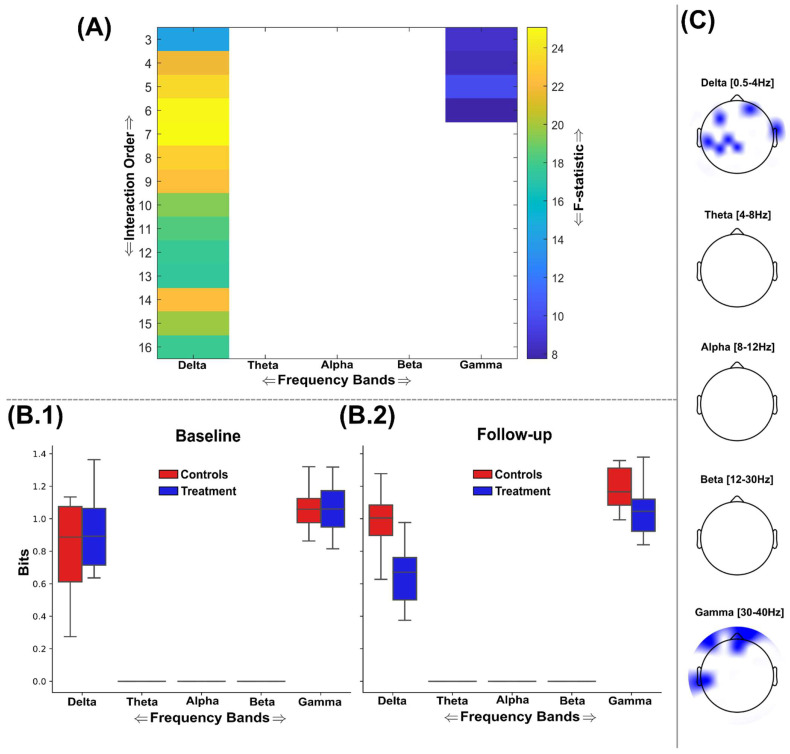
An overview of the findings from the application of O-Information (Ω) to the EEG source signals using the GSA to find maximally discriminative networks of synergistic interactions. (**A**) The F-statistic representing the group differences for each frequency band and interaction order taken from ANCOVA models that controlled for baseline differences in Ω values. No significant differences in favour of lower Ω among the control group were found, so the treatment group results are illustrated only. Areas coloured white indicate no significant differences were found. The interquartile ranges of the most significant interaction order Ω values for control (red) and treatment (blue) groups at the baseline (**B.1**) and follow-up (**B.2**) sessions are depicted as boxplots for each frequency band. As the system of EEG sources for participants was strongly redundancy-dominated across sessions, the Ω values illustrated are positive but demonstrate a significant reduction in the treatment cohort (i.e., increased synergy). Boxplots for theta, alpha, and beta bands are not illustrated, as no significant networks were identified within these ranges. The Ω values illustrated were normalised by the interaction order for comparability. (**C**) The most significantly different brain networks were also illustrated topographically, where blue-shaded areas highlighted the included EEG sources.

**Table 1 brainsci-15-00226-t001:** Contents and dosage of Mind Lab Pro supplement per two-capsule serving.

Nutrition Facts	Amount per Serving
Vitamin B6	2.5 mg
Vitamin B9	100 mcg
Vitamin B12	7.5 mcg
Citicoline	250 mg
Bacopa monnieri	150 mg
Organic lion’s mane mushroom	500 mg
Phosphatidylserine	100 mg
N-Acetyl L-Tyrosine	175 mg
L-Theanine	100 mg
Rhodiola rosea	50 mg
Maritime pine bark extract	75 mg

## Data Availability

The data generated from this study are available upon reasonable request. Higher-order brain network dependencies and the greedy search algorithm were quantified using custom Matlab scripts (https://github.com/rubenherzog/high-order-fc-ml (accessed on 30 July 2024)). Bayesian prevalence was employed using open-source codes: https://github.com/robince/Bayesian-prevalence (accessed on the 5 May 2024).

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
