# Peer review of "Effect of a Plant-Based Nootropic Supplement on Perceptual Decision-Making and Brain Network Interdependencies: A Randomised, Double-Blinded, and Placebo-Controlled Study"

_brainsci, 2025, doi:10.3390/brainsci15030226_

Round 1
Reviewer 1 Report
Comments and Suggestions for Authors
O' Reilly et al elaborate on plant-based nootropic supplement on perceptual decision-making and brain network interdependencies, I have the following comments regarding the work:
1. The plant-based nootropic should be discussed extensively on its impact on hormones, as recently the levels of certain hormones were discussed in the context of neurodegeneration affecting cognitive abilities. - Role of orexin in pathogenesis of neurodegenerative parkinsonisms. Neurol Neurochir Pol. 2023;57(4):335-343. doi:10.5603/PJNNS.a2023.0044
2. The effect of nootropics should be analyzed in the context of comorbidities of the patients.
3. The cognitive abilities should be discussed in the context of level of education.
4. The limitations should be discussed more extensively, moreover authors should provide a comprehensive conclusion section.
Author Response
Comment 1: The plant-based nootropic should be discussed extensively on its impact on hormones, as recently the levels of certain hormones were discussed in the context of neurodegeneration affecting cognitive abilities. - Role of orexin in pathogenesis of neurodegenerative parkinsonisms. Neurol Neurochir Pol. 2023;57(4):335-343. doi:10.5603/PJNNS.a2023.0044
Response 1: We thank the reviewer for their consideration of our work. For this point they have raise, we believe it is beyond the scope of the current study which focusses specifically on perceptual decision-making and brain network interdependencies in a healthy and unimpaired cohort using electrophysiology and behavioural measures as none of the results can be directly related to changes in hormones with nootropic supplementation.
Comment 2: The effect of nootropics should be analyzed in the context of comorbidities of the patients.
Response 2:
The cohort analysed in this study were healthy adults with no comorbidities relevant for the execution of the perceptual decision-making task. The inclusion and exclusion criteria for participating in this study included:
“Eligibility screening of participants was guided by the following inclusion criteria: (i) be between 20-59 years old, (ii) be right-handed and (iii) be able to cease taking other dietary supplements for two months. Exclusion criteria included: (i) currently consuming a nootropic supplement, (ii) any known musculoskeletal, or neurological medical conditions or cognitive impairments (iii) have a known diagnosis of epilepsy/history of seizures and (iv) have a known hearing or visual condition that affects daily life function.”
There were no significant differences between groups in terms of age. Hence, we believe that participant comorbidities did not play a significant role in the outcomes of this study and therefore should not be analysed here.
Comment 3:
The cognitive abilities should be discussed in the context of level of education.
Response 3:
The education level was not recorded during this study and therefore cannot be directly addressed in this manuscript. Nevertheless, in the limitations section of the revised manuscript we have pointed anecdotally towards the high educational attainment of the recruited participants as potentially a limitation to the generalisability of the results to the public but suggest that this is likely to be very minor as this experimental paradigm has been employed on various groups, e.g. young and old aged groups.
“Moreover, we found anecdotally that the participants recruited had a disproportionately high-level of educational attainment compared to the general population, suggesting their cognitive function and so may not be fully representative of the general population. Nonetheless, the cognitive load required for this task was low and previous studies employing this experimental paradigm have found it to be generally applicable to various populations (30).”
Comment 4: The limitations should be discussed more extensively, moreover authors should provide a comprehensive conclusion section.
Response 4:
We have added additional text to both the discussion and limitations sections highlighting important considerations when interpreting the results.
We have also included a separate conclusion section in the revised manuscript:
“Conclusion
This study provides neurocomputational insight into the discrepancy between the growing popularity of natural nootropic supplements in the healthy adult population and their inconclusive cognitive enhancing effects, showing that pronounced neurophysiological effects can occur in the absence of behavioural performance changes. Our findings support the idea that natural nootropic compounds can improve brain health by enhancing the synergy between brain regions, but that these benefits don’t necessarily result in cognitive enhancement among healthy, working age adults, specifically here for rapid visual perceptual decision-making. Future work should focus on more deliberative cognitive tasks and use complementary experimental modalities to comprehensively explain mechanistically the cognitive effects of natural nootropics.”
Reviewer 2 Report
Comments and Suggestions for Authors
The work presented by O’Reilly et al., entitled “Effect of a plant-based nootropic supplement on perceptual decision-making and brain network interdependencies: a randomised, double-blinded and placebo-controlled study” is well-written, clear, and easy to read. The topic is cutting-edge and therefore, it adds further information to the neuroscience area.
In particular, the authors suggest that natural nootropics can improve overall brain network cohesion and energetic efficiency, demonstrating computationally the beneficial effects of natural nootropics on brain health.
The topic is relevant to the field of neurosciences, and the work is congenial
The introduction provides a good overview of nootropics and describes research question and as the authors will respond their hypothesis.
The methods are correct and protocol of randomized control trial is clear (my evaluation is based on NIH tools for RCT. The authors could explain better the pseudo-randomization procedure). The description of EEG preprocessing is thorough.
The results are clearly presented, the figures are in high resolution, but figures could be improved with annotations for significant differences in boxplots.
The conclusions are consistent with the evidence and arguments presented and do they address the main objective. The authors considerer limitation too.
Author Response
Comment 1:
The work presented by O’Reilly et al., entitled “Effect of a plant-based nootropic supplement on perceptual decision-making and brain network interdependencies: a randomised, double-blinded and placebo-controlled study” is well-written, clear, and easy to read. The topic is cutting-edge and therefore, it adds further information to the neuroscience area.
In particular, the authors suggest that natural nootropics can improve overall brain network cohesion and energetic efficiency, demonstrating computationally the beneficial effects of natural nootropics on brain health.
The topic is relevant to the field of neurosciences, and the work is congenial
The introduction provides a good overview of nootropics and describes research question and as the authors will respond their hypothesis.
The methods are correct and protocol of randomized control trial is clear (my evaluation is based on NIH tools for RCT. The authors could explain better the pseudo-randomization procedure. The description of EEG preprocessing is thorough.
The results are clearly presented, the figures are in high resolution, but figures could be improved with annotations for significant differences in boxplots.
The conclusions are consistent with the evidence and arguments presented and do they address the main objective. The authors considerer limitation too.
Response 1:
We thank the reviewer for their positive comments.
Reviewer 3 Report
Comments and Suggestions for Authors
The study follows a randomized, double-blinded, placebo-controlled methodology, which is the gold standard for intervention trials. The application of information-theoretic measures and a greedy search algorithm (GSA) for analyzing brain network interdependencies is an advanced and relatively novel approach. The study addresses an important gap in the literature (why subjective nootropic benefits are reported despite inconsistent empirical evidence).
The study began with 37 participants but only 32 completed the study (17 in the treatment group, 15 in the control group). Given the high inter-individual variability in cognitive function and EEG measures, this sample is underpowered to detect subtle behavioral effects. A larger sample size (e.g., N > 50 per group) would increase statistical power.
The study applies multiple statistical analyses to EEG data across different frequency bands and interaction orders. While Benjamini-Hochberg FDR correction is used, the sheer number of tests increases the risk of false positives. Adjust for multiple comparisons using Bonferroni correction or apply permutation-based statistics to ensure robustness.
The perceptual decision-making task (face/car classification) may not be sufficiently sensitive to detect cognitive benefits from nootropics. Reaction times and accuracy were already high (~80-90%) at baseline, making it difficult to observe improvements. Include more cognitively demanding tasks (e.g., working memory, creativity tests) or assess performance under cognitive fatigue or sleep deprivation where nootropics may have more pronounced effects.
60 days may be insufficient to observe significant behavioral effects. Prior research suggests that chronic neuroplasticity changes might require longer durations (e.g., 3-6 months). Future studies should use a longer intervention period or conduct longitudinal follow-ups.
The study reports increased brain network interdependencies (synergy and redundancy) post-supplementation. However, the choice of EEG sources for network analysis is not fully justified (how were the most relevant channels selected?) Report pre-registered electrode selection criteria or perform whole-brain connectivity analyses.
The study infers cognitive enhancement from increased synergistic interactions (DTC) and redundant interactions (TC). However, synergy and redundancy do not directly imply cognitive enhancement. They could also reflect compensatory or maladaptive changes. Correlate EEG changes with subjective reports or other cognitive measures to confirm their functional significance.
The mechanisms behind the observed EEG changes are not explored. No measures of neurotransmitters (e.g., dopamine, acetylcholine) or blood flow (e.g., fMRI, cerebral blood oxygenation) were included. Future studies should include biomarkers (e.g., fMRI, metabolic measures) to establish causal mechanisms.
The conclusion states that the study explains why nootropics are popular despite weak behavioral evidence. However, no direct measures of subjective well-being were collected. Include self-reported cognitive benefits (e.g., subjective alertness, productivity).
The funding source (Performance Lab Group) is a potential conflict of interest. While the authors declare no direct influence, industry-funded trials have a known bias toward positive findings. Increase transparency by pre-registering analysis plans and hypotheses.
Author Response
Comment 1:
The study follows a randomized, double-blinded, placebo-controlled methodology, which is the gold standard for intervention trials. The application of information-theoretic measures and a greedy search algorithm (GSA) for analysing brain network interdependencies is an advanced and relatively novel approach. The study addresses an important gap in the literature (why subjective nootropic benefits are reported despite inconsistent empirical evidence).
The study began with 37 participants but only 32 completed the study (17 in the treatment group, 15 in the control group). Given the high inter-individual variability in cognitive function and EEG measures, this sample is underpowered to detect subtle behavioral effects. A larger sample size (e.g., N > 50 per group) would increase statistical power.
Response 1:
We agree with the reviewer that a sufficient number of participants is necessary for appropriate testing of group differences, especially for small effects. In the revised manuscript, we have included additional statistical testing at the within-participant level to capitalise on the large number of trials recorded (~700 trials per session) for examining effects on reaction times. We did so using a Bayesian prevalence approach which enables the inference of group-level effects from these individual-level statistics. We found that this approach did not demonstrate any significant group effect as an equivalent minority of each group demonstrated significant improvements in reaction times. The changes made to the manuscript to reflect this change include at the statistical analysis section of the methods and the results:
“To ensure our results are not limited by a sample size, we also capitalised on the large number of trials of reaction-time data to test for statistical differences between pre- and post-sessions at the within participant-level using a Bayesian prevalence approach (47). Specifically, test statistics were generated for each participant and group-level statistical differences were inferred through Monte Carlo simulations.”
“Further testing of these differences at the within participant-level using a Bayesian prevalence approach revealed no group-level effects could be inferred, with a minority of both groups (treatment=7, controls=5) demonstrating significant reductions in choice reaction times.”
“The conclusions drawn from this study’s behavioural findings are limited by the sample size included, although within-participant statistical testing supported our findings and the fact the numbers are in line with other studies, this should still be considered.”
Ince RA, Paton AT, Kay JW, Schyns PG. Bayesian inference of population prevalence. Elife. 2021 Oct 6;10:e62461.
Comment 2:
The study applies multiple statistical analyses to EEG data across different frequency bands and interaction orders. While Benjamini-Hochberg FDR correction is used, the sheer number of tests increases the risk of false positives. Adjust for multiple comparisons using Bonferroni correction or apply permutation-based statistics to ensure robustness.
Response 2:
The statistical testing performed in this study consisted of multiple separate ANCOVA models, with each model corresponding to a unique hypothesis. Hence, the accumulation of family-wise error is not of concern here as the models are not mutually dependent in rejecting the null hypothesis. The relevant literature supports this, stating that “Alpha adjustment is inappropriate in the case of individual testing, in which each individual result must be significant in order to reject each associated individual null hypothesis.” (Rubin, 2021). The testing performed in this study is done on an individual basis, analysing each frequency-specific network at each interaction order separately between pre- and post-sessions. In this way, our focus on reducing the false discovery rate (rather than multiple comparisons) is more applicable here and is supported by the relevant literature.
Rubin, M., 2021. When to adjust alpha during multiple testing: a consideration of disjunction, conjunction, and individual testing. Synthese 199, 10969–11000. https://doi.org/10.1007/s11229-021-03276-4
Comment 3:
The perceptual decision-making task (face/car classification) may not be sufficiently sensitive to detect cognitive benefits from nootropics. Reaction times and accuracy were already high (~80-90%) at baseline, making it difficult to observe improvements. Include more cognitively demanding tasks (e.g., working memory, creativity tests) or assess performance under cognitive fatigue or sleep deprivation where nootropics may have more pronounced effects.
Response 3:
We acknowledge that the participants in this study demonstrated high choice accuracies, suggesting the task may not have been sufficiently difficult for some participants. However, for the EEG analysis, this low error rate provided us with a large number of valid trials to analyse which ultimately lead to the main take-away results of this paper. This has been directly mentioned in the discussion section of the revised manuscript here:
“The participants of both groups exhibited high choice accuracies at both sessions, which could be perceived as the task not being sufficiently difficult to tease out the treatment effects. However, this low error rate provided us with a large number of valid trials to probe the brain networks underpinning perceptual decision-making, which ultimately lead to identifying consistent enhancement of functional complementary across brain regions in the treatment group”
The high-functioning cohort recruited here was also mentioned as a limitation in the revised manuscript:
“Moreover, we found anecdotally that the participants recruited had a disproportionately high-level of educational attainment compared to the general population, suggesting their cognitive function and so may not be fully representative of the general population. Nonetheless, the cognitive load required for this task was low and previous studies employing this experimental paradigm have found it to be generally applicable to various populations (e.g. young versus old) (30).”
Finally, the face-car experimental paradigm has been widely used in perceptual decision-making research, leading to novel scientific insights. Applying this paradigm here to assess nootropic effects demonstrated how pronounced neural effects can coincide with no tangible enhancement in behavioral outcomes.
Comment 4:
60 days may be insufficient to observe significant behavioral effects. Prior research suggests that chronic neuroplasticity changes might require longer durations (e.g., 3-6 months). Future studies should use a longer intervention period or conduct longitudinal follow-ups.
Response 4:
It should be noted that nootropic effects have been evidenced from supplementation for similar durations (see references below), the supplementation duration was decided based on a combination of project constraints and the risk of loss of adherence to the supplementation regime among participants. The decided duration was cited as a potential source of study limitation in the original submission and in the revised manuscript.
Wightman EL, Jackson PA, Khan J, Forster J, Heiner F, Feistel B, Suarez CG, Pischel I, Kennedy DO. The acute and chronic cognitive and cerebral blood flow effects of a Sideritis scardica (Greek mountain tea) extract: A double blind, randomized, placebo controlled, parallel groups study in healthy humans. Nutrients. 2018 Jul 24;10(8):955.
Solomon TM, Leech J, deBros GB, Murphy CA, Budson AE, Vassey EA, Solomon PR. A randomized, double‐blind, placebo controlled, parallel group, efficacy study of alpha BRAIN® administered orally. Human Psychopharmacology: Clinical and Experimental. 2016 Mar;31(2):135-43.
Barringer N, Crombie A, Kotwal R. Impact of a purported nootropic supplementation on measures of mood, stress, and marksmanship performance in US active duty soldiers. Journal of the International Society of Sports Nutrition. 2018 Dec;15:1-6.
Comment 5:
The study reports increased brain network interdependencies (synergy and redundancy) post-supplementation. However, the choice of EEG sources for network analysis is not fully justified (how were the most relevant channels selected?) Report pre-registered electrode selection criteria or perform whole-brain connectivity analyses.
Response 5:
No arbitrary choices were made about EEG sources in this study. Instead, we implemented a data-driven approach (i.e. the greedy-search algorithm) on all available EEG channels to uncover the set of EEG sources that were most significantly different between groups. The code for this has been made available here: https://github.com/rubenherzog/high-order-fc-ml and has been implemented in previous research1.
- Herzog R, Rosas FE, Whelan R, Fittipaldi S, Santamaria-Garcia H, Cruzat J, Birba A, Moguilner S, Tagliazucchi E, Prado P, Ibanez A. Genuine high-order interactions in brain networks and neurodegeneration. Neurobiology of Disease. 2022 Dec 1;175:105918.
Comment 6:
The study infers cognitive enhancement from increased synergistic interactions (DTC) and redundant interactions (TC). However, synergy and redundancy do not directly imply cognitive enhancement. They could also reflect compensatory or maladaptive changes. Correlate EEG changes with subjective reports or other cognitive measures to confirm their functional significance.
Response 6:
The main take-away from this manuscript is that while no observable changes in performance are found, major neurological changes can occur concurrently in a healthy adult cohort. We did not, in fact, link the changes in redundancy and synergy with cognitive enhancement as our findings do not support this link. Instead, we link this neurophysiological finding with literature showing an enhanced synergy/redundancy balance favouring synergy is beneficial towards overall brain health, which doesn’t mean cognitive enhancement. We also provide evidence that this enhancement in synergy is a sustainable change in brain network functioning, with compensatory increases in redundancy to maintain their crucial balance, which in the case of impairment would not be present. We are aware of the connection between hyper functional connectivity and maladaptive brain changes (which we view as an enhanced redundancy at the expense of synergy in contrast), however, we are unaware of any literature suggesting enhanced synergy between brain regions as being a maladaptive response in healthy adults and have provided several references below here supporting our position:
Gatica M, Cofré R, Mediano PA, Rosas FE, Orio P, Diez I, Swinnen SP, Cortes JM. High-order interdependencies in the aging brain. Brain connectivity. 2021 Nov 1;11(9):734-44.
Proca AM, Rosas FE, Luppi AI, Bor D, Crosby M, Mediano PA. Synergistic information supports modality integration and flexible learning in neural networks solving multiple tasks. PLOS Computational Biology. 2024 Jun 3;20(6):e1012178.
Brenner N, Strong SP, Koberle R, Bialek W, Steveninck RR. Synergy in a neural code. Neural computation. 2000 Jul 1;12(7):1531-52.
Nigam S, Pojoga S, Dragoi V. Synergistic coding of visual information in columnar networks. Neuron. 2019 Oct 23;104(2):402-11.
Luppi AI, Rosas FE, Mediano PA, Menon DK, Stamatakis EA. Information decomposition and the informational architecture of the brain. Trends in Cognitive Sciences. 2024 Jan 9.
Combrisson E, Basanisi R, Neri M, Auzias G, Petri G, Marinazzo D, Panzeri S, Brovelli A. Higher-order and distributed synergistic functional interactions encode information gain in goal-directed learning. bioRxiv. 2024:2024-09.
Comment 7:
The mechanisms behind the observed EEG changes are not explored. No measures of neurotransmitters (e.g., dopamine, acetylcholine) or blood flow (e.g., fMRI, cerebral blood oxygenation) were included. Future studies should include biomarkers (e.g., fMRI, metabolic measures) to establish causal mechanisms.
Response 7:
The use of complementary experimental modalities in future work has been mentioned in the revised manuscript:
“Future research on these compounds in the healthy adult population should therefore examine their effects using complementary modalities (e.g. fMRI) across greater supplementation durations and in more deliberate cognitive tasks.”
Comment 8:
The conclusion states that the study explains why nootropics are popular despite weak behavioral evidence. However, no direct measures of subjective well-being were collected. Include self-reported cognitive benefits (e.g., subjective alertness, productivity).
Response 8:
Apologies for any misunderstanding on this but no defining statement was made in the discussion section suggesting we have answered this question regarding the popularity of nootropics despite weak behavioural outcomes. Instead, we provided evidence for enhanced synergy in the brain following supplementation which is linked with various factors related to brain health. We then go on to suggest future directions that can more comprehensively answer this question. Subjective well-being reports were not collected in this study and are therefore beyond the studys current scope.
In the revised manuscript, we have included a conclusion section which more explicitly states our position in this regard:
“Conclusion
This study provides neurocomputational insight into the discrepancy between the growing popularity of natural nootropic supplements in the healthy adult population and their inconclusive cognitive enhancing effects, showing that pronounced neurophysiological effects can occur in the absence of behavioural performance changes. Our findings support the idea that natural nootropic compounds can improve brain health by enhancing the synergy between brain regions, but that these benefits don’t necessarily result in cognitive enhancement among healthy, working age adults, specifically here for rapid visual perceptual decision-making. Future work should focus on more deliberative cognitive tasks and use complementary experimental modalities to comprehensively explain mechanistically the cognitive effects of natural nootropics.”
Comment 9:
The funding source (Performance Lab Group) is a potential conflict of interest. While the authors declare no direct influence, industry-funded trials have a known bias toward positive findings. Increase transparency by pre-registering analysis plans and hypotheses.
Response 9:
Our findings are not all positive, including non-significant behavioural results, and in fact highlight important uncertainties in the utility of nootropic supplements as cognitive enhancers (which the company has advertised their product as) in the general adult population. A pre-registration of the study was provided in the original manuscript (clinical trials registration number: NCT06689644 and https://osf.io/25afe). Moreover, the large-scale data we have generated during this experiment is available upon reasonable request to the authors. Finally, this study was approved by an ethics board independently of the funding company that would have assessed the risk of bias in the project.
Round 2
Reviewer 1 Report
Comments and Suggestions for Authors
Authors revised most of the points indicated on the first round of review, however I believe that the point: "The plant-based nootropic should be discussed extensively on its impact on hormones, as recently the levels of certain hormones were discussed in the context of neurodegeneration affecting cognitive abilities" should be adequately addressed.
Author Response
Comment 1: Authors revised most of the points indicated on the first round of review, however I believe that the point: "The plant-based nootropic should be discussed extensively on its impact on hormones, as recently the levels of certain hormones were discussed in the context of neurodegeneration affecting cognitive abilities" should be adequately addressed.
Response 1:
We thank the reviewer for their consideration of our study. In the latest revision of this work, we have included specific examples of the impact of natural nootropics on neurohormones among other effects in the introduction:
"The mechanisms by which these nootropics restore cognitive abilities varies widely but commonly involves increasing the supply and efficiency of use of energetic resources in the brain (e.g. increased cerebral blood flow, positive allosteric modulation of acetylcholine and glutamate receptors and inhibition of monoamine oxidases) (8,12–15). Additionally, they have been shown to promote neurogenesis, stimulate phospholipid metabolism in neurohormonal membranes and enhance the beneficial cognitive effects of neurosteroids (e.g. pregnenolone sulfate, dehydroepiandrosterone sulfate) (12,16–19). "
With this additional text, we believe readers will now have a comprehensive awareness of the widespread effects of nootropics on brain function, including the significant impact of neurohormones on cognition and their relationship with these compounds.